# Green Tea Seed Oil Suppressed Aβ_1–42_-Induced Behavioral and Cognitive Deficit via the Aβ-Related Akt Pathway

**DOI:** 10.3390/ijms20081865

**Published:** 2019-04-15

**Authors:** Jong Min Kim, Seon Kyeong Park, Jin Yong Kang, Su Bin Park, Seul Ki Yoo, Hye Ju Han, Kyoung Hwan Cho, Jong Cheol Kim, Ho Jin Heo

**Affiliations:** 1Division of Applied Life Science (BK21 plus), Institute of Agriculture and Life Science, Gyeongsang National University, Jinju 52828, Korea; myrock201@naver.com (J.M.K.); tjsrud2510@naver.com (S.K.P.); kangjy2132@naver.com (J.Y.K.); tbsk5670@naver.com (S.B.P.); ysyk9412@naver.com (S.K.Y.); gksgpwn2527@naver.com (H.J.H.); 2Institute of Hadong Green Tea, Hadong 52304, Korea; ckh8568@hgreent.or.kr (K.H.C.); jckim@hgreent.or.kr (J.C.K.)

**Keywords:** green tea seed oil, amyloid β, neuroprotective effect, Aβ-related Akt pathway

## Abstract

The aim of this study was to investigate the availability of seeds, one of the byproducts of green tea, and evaluate the physiological activity of seed oil. The ameliorating effect of green tea seed oil (GTO) was evaluated on H_2_O_2_-induced PC12 cells and amyloid beta (Aβ)_1–42_-induced ICR mice. GTO showed improvement of cell viability and reduced reactive oxygen species (ROS) production in H_2_O_2_-induced PC12 cells by conducting the 2′,3-(4,5-dimethylthiazol-2-yl)-2,5-diphenyl tetrazolium bromide (MTT) and 2′,7′-dichlorofluorescein diacetate (DCF-DA) analysis. Also, administration of GTO (50 and 100 mg/kg body weight) presented protective effects on behavioral and memory dysfunction by conducting Y-maze, passive avoidance, and Morris water maze tests in Aβ-induced ICR mice. GTO protected the antioxidant system by reducing malondialdehyde (MDA) levels, and by increasing superoxide dismutase (SOD) and reducing glutathione (GSH) contents. It significantly regulated the cholinergic system of acetylcholine (ACh) contents, acetylcholinesterase (AChE) activities, and AChE expression. Also, mitochondrial function was improved through the reduced production of ROS and damage of mitochondrial membrane potential (MMP) by regulating the Aβ-related c-Jun N-terminal kinase (JNK)/protein kinase B (Akt) and Akt/apoptosis pathways. This study suggested that GTO may have an ameliorating effect on cognitive dysfunction and neurotoxicity through various physiological activities.

## 1. Introduction

Alzheimer’s disease (AD) is a serious neurodegenerative disease that gradually causes loss of memory and cognition. AD is a disease that afflicts patients, families, and society, and is characterized by motor dysfunction and cognitive impairment [1]. The mechanism related to neuronal death in AD has not yet been completely ascertained, but excessive oxidative stress and inflammatory toxicity are regarded as major initiators or mediators of AD [2]. Also, the pathological features of AD are closely related to the loss of cerebral cholinergic neurons and significance of acetylcholine levels in AD [3]. One of the pathological features in AD patients is senile plaques produced by various causes. Senile plaques consist of various Aβ peptides containing the Aβ_1–40_ and Aβ_1–42_ amino acid cleavage products of amyloid precursor protein (APP), one of the most abundant proteins, as transmembrane proteins, and intracellular neurofibrillar tangles (NFTs), such as hyper-phosphorylated tau (p-tau) protein [4]. In AD patients, APP is abnormally degraded to Aβ by β- and γ-secretase and decomposed to Aβ. There are several Aβ peptides in the form of Aβ_1–40_ and Aβ_1–42_, which are most commonly found in the brains of AD patients [2]. Aβ is excreted as a form of monomer, and releases Aβ aggregates into protofibrils and fibrils to form mature amyloid plaques [4]. In particular, Aβ_1–42_ is prone to aggregation and fibrosis. Aβ_1–42_ plays an important role in the pathogenesis of AD because it is the most toxic compared to other species. Aggregated Aβ interacts with neuronal and glial cells, and this reaction causes the activation of inflammatory responses, production of oxidative stress, hyperphosphorylation of tau protein, and neuronal apoptosis [5]. These undesirable phenomena ultimately cause cognitive deficits, behavior and memory dysfunctions, and psychiatric symptoms [6].

To prevent these diseases, the ingestion of nutraceuticals and antioxidants has been reported to be important. These are physiologically active substances that originate from plants and animals and help to alleviate pathological conditions [7]. They also have the ability to prevent disease by inhibiting the onset of pathological conditions, or alleviating the progression of the disease [8]. Nutraceuticals include dietary fiber, probiotics, prebiotics, polyunsaturated fatty acids, antioxidant vitamins, polyphenols, and spices, which are known to have antioxidant and anti-inflammatory properties [9].

Green tea (*Camellia sinensis*), one of these nutraceuticals, contains various antioxidants such as catechins, which include epigallocatechin gallate (EGCG), epicatechin gallate (ECG), epicatechin (EC), and epigallocatechin (EGC) and have an excellent effect on chronic diseases such as cardiovascular diseases [10]. The content of catechin is different according to the cultivation area, but it generally shows higher contents in the order of EGCG, EGC, ECG, and EC [11]. Green tea has physiological effects such as lowering plasma cholesterol, and the inhibition of lipid peroxidation and hepatic triglyceride accumulation [12]. Also, green tea has potential neuroprotective effects for various brain injuries such as stroke and AD [13]. Green tea seed contains three times more saponin and oleic acid than soybeans. In addition, green tea seed has many physiologically active substances such as oils, flavonoids, vitamins, polyphenols, and saponins [14]. The consumption of green tea is increasing, and the production of seeds is also increasing. However, tea seeds are almost always discarded after harvest because they are not used to make oil, even though they contain a variety of physiologically active substances [15]. Therefore, the major aim of this study concerning the evaluation of availability of green tea oil, which is one of the byproducts of green tea, was to investigate the neuronal protective effect in PC12 cells and cognitive dysfunction-induced mice.

## 2. Results

### 2.1. Identification of Main Compounds Using UPLC Q-TOF/MS^2^

The major physiological compounds of green tea seed oil (GTO) were qualitatively identified using ultra-performance liquid chromatography-ion mobility separation-quadrupole time of flight/tandem mass spectrometry (UPLC IMS Q-TOF/MS^2^) analysis (Figure 1 and Table 1). The MS spectra (Figure 1) were continuously obtained in negative ion mode [M−H]^−^ as compound A, 741 m/z (RT: 3.31 min); compound B, 565 *m*/*z* (RT: 3.33 min); compound C, 271 *m*/*z* (RT: 4.11 min); compound D, 329 m/z (RT: 4.29 min); compound E, 517 *m*/*z* (RT: 4.50 min) and compound F, 501 *m*/*z* (RT: 4.95 min). When the main fragments were compared with the previous study, these peaks were identified as narirutin 4′-glucoside (compound A, PubChem CID: 14375140), quercetin-3-*O*-pentosyl-pentoside (compound B, PubChem CID: 45360047), naringenin (compound C, PubChem CID: 932), 9,12,13-trihydroxy-10-octadecenoic (9,12,13-THODE) acid (compound D, PubChem CID: 129723910), ganoderic acid C2 (compound E, PubChem CID: 57396771) and ganolucidic acid B (compound F, PubChem CID: 20055994), respectively [16,17,18,19,20,21].

### 2.2. Cell Viability and Intracellular ROS

The neuroprotective effect of GTO against H_2_O_2_-induced neurotoxicity (Figure 2A), showed that the H_2_O_2_-treated group had low cell viability (68.94%) compared to the control group (100%). However, the viability of the GTO-treated groups (2000 μg/mL, 79.63%; 1000 μg/mL, 75.22%) increased compared to the H_2_O_2_-treated group.

A DCF-DA assay was conducted to evaluate the ameliorating effect of GTO on H_2_O_2_-induced intracellular ROS production in PC12 cells (Figure 2B). The H_2_O_2_-treated group (129.14%) presented increased reactive oxygen stress (ROS) amounts compared to that of the control group (100%), whereas the GTO-treated groups (2000 μg/mL, 115.23%; 1000 μg/mL, 123.33%) showed significantly reduced intracellular ROS.

### 2.3. Y-Maze Test

To assess spatial learning and memory function, a Y-maze test was conducted (Figure 3). The Aβ group showed a significant reduction in spontaneous alternation behavior (44.49%) compared to the NC group (61.41%). The alternation behaviors of the GTO groups (49.85% and 51.11%; GTO 50 and GTO 100, respectively) were improved compared to the Aβ group (Figure 3A), while the number of arm entries of all the groups were not statistically different (Figure 3B). This indicated that the Aβ injection did not affect the physical activity of the mice. In the results of path tracing (Figure 3C), the Aβ group showed irregular movements compared to the NC group, but the GTO 100 group was similar to the NC group.

### 2.4. Passive Avoidance Test

Short-term memory ability was measured using a passive avoidance test (Figure 4). The first step-through latency showed no significant differences between all the groups (Figure 4A). However, in the trial test (Figure 4B), the Aβ group (168.67 s) showed a reduction in latency time in compared with the NC group (296.67 s). However, the GTO groups’ times were 200.50 s and 258.33 s (GTO 50 and GTO 100, respectively).

### 2.5. Morris Water Maze Test

To measure spatial learning acquisition and long-term memory, a Morris water maze test was performed (Figure 5). In the hidden trial (Figure 5A), escape latency for the platform gradually decreased. In the hidden trial, the escape latency of the Aβ group (43.82 s) decreased less than the NC group (33.27 s), while that of the GTO groups (39.38 s and 36.66 s; GTO 50 and GTO 100, respectively) decreased more than the Aβ group. In the probe test (Figure 5B), the retention time in the W zone for the Aβ group (44.49%) decreased compared to that of the NC group (58.64%). However, that of the GTO groups (49.85% and 54.11%; GTO 50 and GTO 100, respectively) increased compared to the Aβ group. Comparing this with recorded movements, the movements of the Aβ group were less animated than those of the NC group (Figure 5C). On the other hand, the GTO groups showed improved movement compared to the Aβ group.

### 2.6. Antioxidant System in Brain Tissue

To confirm the protective effect of the antioxidant system, malondialdehyde (MDA) levels, superoxide dismutase (SOD) contents and reduced glutathione (GSH) contents were measured (Table 2). The MDA level of the Aβ group (54.52 ± 5.03 nmol/mg of protein) increased compared to the (54.52 nmol/mg of protein) increased compared to the NC group (40.88 nmol/mg of protein). The MDA level in the GTO groups (47.62 nmol/mg of protein and 44.93 nmol/mg of protein; GTO 50 and GTO 100, respectively) decreased compared to that of the Aβ group. The SOD contents of the Aβ group (30.73 U/mg of protein) were lower compared to the NC group (44.18 U/mg of protein). The GTO groups (35.99 U/mg of protein and 38.85 U/mg of protein; GTO 50 and GTO 100, respectively) were increased compared to the Aβ group. The reduced GSH contents of Aβ group (2.02 μg GSH/mg of protein) was lower compared to that of the NC group (2.79 μg GSH/mg of protein). The GTO groups (2.14 μg GSH/mg of protein and 3.22 μg GSH/mg of protein; GTO 50 and GTO 100, respectively) were ameliorated compared to the Aβ group.

### 2.7. Cholinergic System in Brain Tissue

To examine the ameliorating effect of GTO on Aβ-induced cholinergic dysfunction, acetylcholine (ACh) contents and acetylcholinesterase (AChE) activity were examined (Figure 6). The ACh contents are shown in Figure 6A. The ACh contents of the Aβ group (2.81 ± 0.13 mmol/mg of protein) decreased compared to the NC group (3.52 ± 0.47 mmol/mg of protein), while that of the GTO groups (3.00 ± 0.49 mmol/mg of protein and 3.09 ± 0.75 mmol/mg of protein) showed a significant increase compared to the Aβ group. The activities of AChE are shown in Figure 6B. The AChE of the Aβ group (129.23 ± 10.53%) was activated more compared to the NC group (100 ± 11.99%). On the other hand, that of the GTO groups (121.29 ± 12.29% and 111.32 ± 10.60%; GTO 50 and GTO 100, respectively) were less activated compared to the Aβ group. AChE expression is presented in Figure 6C,D. The AChE expression level in the Aβ group was considerably upregulated by 19.26 ± 3.10% compared to the NC group. Treatment with GTO considerably downregulated AChE expression (20.49 ± 3.05%) compared to the Aβ group.

### 2.8. Mitochondrial Function in Brain Tissue

To assess the ameliorating effect of GTO on Aβ-injected mitochondrial dysfunction, ROS production, MMP, and ATP levels were investigated (Table 3). The Aβ group (140.41 ± 20.24%) showed increased DFC formation compared to that of the NC group (100 ± 19.83%), whereas the GTO groups (130.52 ± 2.88% and 106.47 ± 13.36%; GTO 50 and GTO 100, respectively) showed considerably decreased DCF production compared with the Aβ group. The mitochondrial membrane potential (MMP) of the Aβ group (68.74 ± 6.17%) declined in comparison with the NC group (100 ± 5.12%), while both GTO groups (79.02 ± 7.31% and 80.04 ± 16.16%; GTO 50 and GTO 100, respectively) exhibited remarkably improved membrane potential compared to the Aβ group.

### 2.9. Protein Expression via the Aβ-Related JNK/Akt Pathway

To evaluate the regulating effect of GTO on Aβ-induced ICR mice via the Aβ-related JNK/Akt pathway, the protein expressions of TNF-α, p-JNK, p-Akt, p-tau and p-NF-κB were measured (Figure 7). The expression levels of TNF-α and p-JNK in the Aβ group were significantly upregulated, by 28.85 ± 1.60% and 35.87 ± 2.24%, respectively, compared to the NC group (Figure 7B,C). Whereas the GTO group downregulated TNF-α and p-JNK expressions levels by 22.28 ± 2.73% and 10.26 ± 0.72%, respectively, compared to the Aβ group. The expression level of p-Akt in the Aβ group was significantly downregulated by 25.09 ± 6.98% compared to the NC group (Figure 7D), while the GTO groups upregulated the p-Akt expression level by 22.57 ± 4.91% compared to the Aβ group. The expression levels of p-tau and p-NF-κB in the Aβ group were considerably upregulated, by 33.94 ± 8.77% and 43.73 ± 6.34%, respectively, compared to the NC group (Figure 7E,F). On the other hand, the GTO group downregulated p-tau and p-NF-κB expression levels by 22.14 ± 8.33% and 24.76 ± 7.73%, respectively, compared to the Aβ group.

### 2.10. Protein Expression via the Aβ-Related Akt/Apoptosis Pathway

To evaluate the regulating effect of GTO on Aβ-induced ICR mice via the Aβ-related Akt/apoptosis pathway, the protein expressions of BAX, mitochondrial and cytosolic cytochrome c were measured (Figure 8). The expression levels of BAX and cytosolic cytochrome c in the Aβ group were significantly upregulated, by 23.18 ± 0.88% and 31.95 ± 4.95%, respectively, compared to the NC group (Figure 8B,C). However, the GTO group downregulated BAX and cytosolic cytochrome c expressions levels by 46.17 ± 7.55% and 37.30 ± 19.16%, respectively, compared to the Aβ group. The expression level of mitochondrial cytochrome c in the Aβ group was significantly downregulated by 40.21 ± 2.98% compared to the NC group (Figure 8D), but the GTO group upregulated the mitochondrial cytochrome c expression level by 54.99 ± 5.47% compared to the Aβ group. Also, the ratio of cytochrome c in cytosol/mitochondria in the Aβ group increased considerably, by 122.62 ± 26.88%, compared to the NC group (Figure 8E), while the GTO group showed a decreased ratio of cytochrome c in cytosol/mitochondria by 58.63 ± 34.19% compared to the Aβ group.

## 3. Discussion

AD is the most general neurodegenerative dementia disease among the elderly. The causes of AD are known to be the aggregation of amyloid beta (Aβ) and the resulting oxidative stress in the neurons. These neuronal cells have been known to be especially sensitive to cellular damage by Aβ. There is evidence that aggregation of Aβ is associated with neurotoxicity, and aggregated Aβ produces free radicals [1]. Aβ leads to H_2_O_2_ accumulation in hippocampal neurons and causes a cascade of AD through apoptosis and neuronal cell death [22]. In particular, it induces lipid peroxidation in hippocampal tissues, oxidative deformation of lipoic acid, and damage of nuclei, mitochondria, and DNA. The oxidation of neuronal cell by Aβ results in impaired cognitive function, and ultimately causes AD [23]. Therefore, we investigated whether the consumption of GTO can regulate oxidative stress and cognitive dysfunction in an Aβ-induced animal model, including dysfunction of the antioxidant and cholinergic system, and mitochondrial damage.

Neuronal cells in brain tissue are sensitive to oxidative damage due to a deficient antioxidant system and high oxygen demand, although there is a difference in sensitivity to oxidative stress depending on the cell type such as neurons, oligodendrocytes, astrocytes, microglia and macrophages [24]. The brains of AD patients exhibit an increased production of lipid peroxides, such as 4-hydroxynonenal and 2-propenal, and also have oxidative damage to proteins, mitochondria, and nuclear DNA [25]. Therefore, in order to evaluate the protective effect of GTO against oxidative stress, cell viability and ROS production were confirmed in PC12 cells. In these results, this study suggested that GTO reduces oxidative stress and protects PC12 cells (Figure 2). Previous studies have suggested that green tea decreased the amount of ROS and increased the survival rate of Aβ_25–35_-induced PC12 cells [26]. Naringenin, a major polyphenol in green tea and GTO, increased neuronal cell viability and pyruvate amounts, and decreased ROS production and lactate dehydrogenase release in PC12 cells [27]. Similar to the results of green tea, GTO also exhibited protective effects for nerve cells with its excellent neuroprotective activity.

Aβ stimulates oxidative stress and inflammation, causing the death of neurons in the hippocampus, amygdala, thalamus, and cerebellum, and results in cognitive and behavioral impairment. This damage causes cognitive deficits and loss of memory [28]. Therefore, behavioral assessment is one of the most widely used indexes of learning and memory ability and reflects cognitive dysfunction in early stages of AD [1]. The Y-maze test is based on the innate nature of mice to try to enter a new place when exploring a new environment. This study showed that the consumption of GTO improved spatial learning and memory function compared to Aβ-infused mice (Figure 3A). But there was no difference in the movement of the mice, so it seemed that there was no abnormality in their exercise capacity (Figure 3B). Moreover, in step-through latency, the short-term memory ability of the GTO groups was significantly ameliorated in passive avoidance (Figure 4). Also, this research showed that the escape time to the platform of the GTO groups decreased, and GTO increased the time spent in the target zone without a platform in the trial (Figure 5). When compared with Aβ-induced mice, GTO improved the spatial memory function, short-term memory and spatial learning ability. Previous studies have reported that the peel of *Citrus kawachiensis*, which contains narirutin, ameliorates cognitive dysfunction by regulating the suppression of microglial activation in the hippocampus, decreasing the phosphorylation of tau and protecting neuronal synapses [29]. Also, the peel of *Citrus reticulata* containing narirutin downregulated inflammatory markers such as inflammation-related genes such as nitric oxide synthase (iNOS) and cyclooxygenase-2 (COX-2) protein and mRNA expression-related cognitive dysfunction [30]. Methanol extract of *Zizania latifolia*, which contains a large amount of THODE, showed nitric oxide inhibitory activity and release of β-hexosaminidase, affecting cognitive deficit on lipopolysaccharide (LPS)-induced production of RAW 264.7 cells, and IgE-sensitized RBL-2H3 cells [21,31]. Based on these results, it is suggested that GTO has a protective effect on cognitive and memory functions in Aβ-induced mice.

The level of inflammatory response, ROS production and oxidative stress markers in the brain of AD patients is elevated. On the other hand, levels of antioxidant enzymes have been reported to decrease [22]. In particular, lipid peroxidation accumulates in the brain due to oxidative stress and exists at a high level. Oxidative stress in AD patients is thought to be related to amyloid pathology [23]. In this study, some biomarkers such as MDA, SOD, and reduced GSH were measured to determine whether GTO reduces oxidative stress and improves the antioxidant system in the brain tissue of Aβ-induced mice. GTO regulated the antioxidant system by inhibiting MDA production and increasing SOD and reduced GSH contents. Naringenin in GTO, considerably reduced MDA and nitrite content, and increased SOD activity in the hippocampus of Aβ_1–40_-induced mouse models [32]. Similar to GTO, *Laurus nobilis*, which contains a large amount of quercetin-3-O-pentosyl-pentoside, has been reported to reduce DNA fragmentation and reduce oxidative stress to protect nerve cells [33]. Based on these results, it is suggested that the protective effect of the antioxidant system in brain tissue is due to various phenolic compounds of GTO (Table 2). Through the protective effect of GTO, progress of antioxidant system dysfunction was inhibited, and this is expected to eventually be able to prevent cognitive decline.

The cholinergic system plays an important role in cognitive function. The death of cholinergic cells by the toxicity of Aβ increases the release of AChE bound to cells, and the increased AChE accelerates the degradation of ACh in the synaptic cleft [34]. The continuously increased production of Aβ fibrils in the hippocampus promotes the formation of Aβ-AChE complexes. The toxicity of this complex is known to be greater than the toxicity of common Aβ. Increased Aβ-induced AChE causes damage to the cholinergic system and finally causes neuronal death [35]. Therefore, the improvement effect on the cholinergic system was confirmed to measure the ACh content and AChE activity in the brain of Aβ-injected mice fed GTO (Figure 6). The amount of ACh in the mouse brain of the GTO group increased, and the activity of AChE was significantly decreased compared to the Aβ group. In addition, *Ganoderma lucidum*, which contains the ganoderic acid C2, had a significant AChE inhibitory effect [32]. The presence of ganoderic acid C2 in rat plasma was stable in the original form or through minor hydroxylation, dehydrogenation and oxidative metabolites that were structurally identified by HPLC–DAD–ESI-MS^n^ and LC–ESI-IT-TOF/MS [23]. In addition, the cholinergic system is closely related to the antioxidant system, and AChE is fixed on the cell membrane’s surface. However, if cell membrane damage due to oxidative stress persists, AChE is released into the cytoplasm by lipid peroxidation. The released AChE promotes the acceleration of ACh decomposition, and this results in a decline in cognitive function [36]. Similarly, broccoli (*Brassica oleracea* var. *italica*) leaves, which contain THODE, inhibited AChE activity by reducing lipid peroxide in the Aβ_1–42_-induced mouse brain [37]. It is suggested that the improvement of cognitive deficit in mice fed GTO seems to be due to the abundance of unsaturated fatty acids such as THODE and various triterpenoid compounds such as ganoderic acid C2. These compounds directly inhibited the activity of AChE to increase the content of ACh, and indirectly decreased the activity of AChE by inhibiting the lipid peroxidation activity of the neuronal membrane. The intake of GTO increased the survival of neurons and decreased the damage of cholinergic system in addition to the improvement of antioxidant system. Therefore, it is suggested that GTO has a protective effect with the physiological activity of phenolic compounds, and has an inhibitory effect on AChE by reducing the lipid peroxidation of neuronal cell membrane.

Aβ produces oxidative stress by binding to the mitochondrial membrane, and this results in mitochondrial damage with increased ROS and lipid peroxidation production [38]. Compared with other organs, the brain is highly susceptible to oxidative stress because of its high content of unsaturated fatty acids, relatively high levels of oxygen metabolism and deficient content of antioxidants [39]. When the mitochondria present in neurons are degraded or lost by oxidative stress or Aβ, the lost mitochondria functions are transported to the synaptic terminals and degraded. Because mitochondria are degraded, the production of ATP is markedly reduced [38]. APP-generated Aβ forms an oligomer at the synaptic terminal. Oligomeric Aβ has a sharp shape and penetrates into cell organelles such as mitochondria [40]. This oligomer damages mitochondria at the synaptic ends, and causes mitochondrial dysfunction [41]. These defects of mitochondrial membranes disrupt synaptic ions and energy metabolism and promote the death of synapses. Sustained damage of the electron transport system causes dysfunction of calcium ion homeostasis and induces MMP disruption [42]. Based on these findings, this study confirmed the effect of improvement of cerebral mitochondria in Aβ-induced mice (Table 3). GTO reduced the amount of ROS produced by Aβ, and improved MMP. In addition, energy metabolism was controlled by inhibiting the decrease of ATP. Naringenin, one of the phenolic compounds of GTO, increased NADH dehydrogenase activity, succinate dehydrogenase activity and mitochondrial viability, and improved mitochondrial enzymes function complex-I and -III by scavenging the ROS in D-galactose-treated mice [43]. High levels of energy metabolism directly increase the activity of neurons and indirectly increase mitochondrial activity, which has been reported to improve cognitive function by increasing the release of hippocampal ACh and level of glial cell line-derived neurotrophic factor [44]. Also, EGCG, which is reported to be abundant in green tea, recovered mitochondrial respiration, MMP, ROS production, and ATP levels in amyloid-induced murine neuroblastoma N2a cells [45]. The physiological activities of catechin compounds such as EGCG with mitochondrial improvement effects are presumed to be similar to these results [46]. Similar to the in vitro protective effect of neurons (Figure 2B), GTO decreased ROS and inhibited damage of neurons and mitochondria. Therefore, mitochondria have a protective effect against oxidative stress due to the influence of these physiologically active substances of GTO. This is related to the energy metabolism, and it may be possible to suppress neurodegeneration by increasing the survival of neuronal cells.

Oligomeric Aβ promotes an increase in cerebral tumor necrosis factor-alpha (TNF-α) levels, an inflammatory necrosis factor, and increases the activity of phosphorylated c-Jun N-terminal kinase (p-JNK), an abnormally activated form of JNK in neurons. p-JNK promotes the phosphorylation of serine residues of IRS-1 (IRS-1pSer) instead of tyrosine residues [47]. IRS-1pSer blocks normal signal transmission and generates an irregular signal, causing the aggregation of Aβ by lowering the activity of the insulin degrading enzyme that inhibits Aβ. Eventually, the abnormal signaling pathway induced by oligomeric Aβ may cause a vicious cycle of constantly upregulating its production and aggregation [48]. Furthermore, this process stimulates the JNK/phosphoinositide 3-kinase (PI3K)/protein kinase B (Akt) pathway and promotes neuronal degeneration by reducing phosphorylated Akt (p-Akt), affecting cell survival and the activation of p53 [49]. Activated p53 increases the ratio of Bcl-2-associated X protein (BAX)/Bcl-2, which induces cytochrome c release from mitochondria into the cytosol. The cytochrome c released by increased BAX activates the expression of caspase-3 and poly [ADP-ribose] polymerase 1 (PARP-1), ultimately leading to apoptosis, and causes the death of nerve cells [47]. Tau is a neuron-related protein that plays an important role in the assembly and stability of microtubules and transport of endoplasmic reticulum in neuronal axons. This tau protein is hyperphosphorylated by a variety of causes. The excessive phosphorylation of tau promotes the accumulation of NFT in the axons of neurons, ultimately leading to damage to synaptic plasticity and cell death [4]. Based on these pathways, we found that GTO significantly reduced the expression of TNF-α, p-JNK, p-tau, p-NF-κB, BAX and cytosolic cytochrome c, which cause neuronal death, and increased the amount of p-Akt and mitochondrial cytochrome c related to neuronal cell survival (Figure 7 and Figure 8). Similar to GTO, *Citrus kawachiensis*, which contains narirutin, depressed the inflammatory indicators as glial fibrillary acidic protein (GFAP) and COX-2 in LPS-induced inflammation in the mouse brain [50]. Also, naringenin significantly regulated the expression of caspase-3, BAX and Bcl-2 in neuronal cells [51]. GTO, which contains various physiologically active substances such as naringenin, regulated JNK/Akt pathway to increase neuronal survival and inhibited apoptosis. GTO showed excellent protective effect of mitochondria, which is thought to suppress the death of neurons by reducing the amount of cytochrome c released from mitochondria (Table 3). These results suggest that GTO significantly inhibits apoptosis-related factors and can prevent neuronal death.

## 4. Materials and Methods

### 4.1. Chemicals

Dimethyl sulfoxide (DMSO), RPMI 1640 medium, penicillin, streptomycin, fetal bovine serum (FBS), 2′,7′-dichlorofluorescein diacetate (DCF-DA), 2′,3-(4,5-dimethylthiazol-2-yl)-2,5-diphenyl tetrazolium bromide (MTT) assay kit, bovine serum albumin, superoxide dismutase (SOD) assay kit, 5,5,6,6-tetrachloro-1,1,3,3-tetraethylbenzimidazolylcarbocyanine iodide (JC-1), sodium hydroxide, hydroxylamine, FeCl_3_, phenylmethanesulfonyl fluoride (PMSF), OPT, mannitol, sucrose, HEPES sodium salt, digitonin, egtazic acid (EGTA), malate, pyruvate, phosphoric acid, metaphosphoric acid, thiobarbituric acid, and solvents were obtained from Sigma-Aldrich Chemical Co. (St. Louis, MO, USA). Aβ_1-42_ was purchased from Bachem (Bubendorf, Switzerland).

### 4.2. Sample Preparation

Green tea seed oil (GTO) was provided by the da’O Agricultural Association Corporation (Hadong, Korea) based on sample information from the Institute of Hadong Green Tea (Hadong, Korea). After separating the outer skin and the inner skin, green tea seed was dried for 24 h using a dryer (DY-22H, Daeyeong E & B, Ansan, Korea). The dried tea was pressed using a low-temperature oil presser (ECPL-0001, National ENG Co., LTD, Goyang, Korea). Pressed samples were stored at 4 °C and protected from the light.

### 4.3. Identification of Main Compounds Using UPLC Q-TOF/MS^2^

To separate and defat the physiologic compounds, the sample was fractionated. Five grams of GTO was vortexed with 10 mL of n-hexane and 12 mL of methanol‒water (60:40, *v*/*v*). The mixture was centrifuged at 3500× *g* for 10 min at 4 °C. The methanol solution was separated, and lyophilized at 40 °C. This extracted sample was dissolved in methanol‒water (50:50, *v*/*v*), and filtered using 0.2 μm organic membrane filters for further analysis. The main physiologic compounds in GTO were analyzed using an ultra-performance liquid chromatography-ion mobility separation-quadrupole time of flight/tandem mass spectrometry (UPLC-IMS-QTOF/MS^2^, Vion, Waters Corp., Milford, MA, USA). UPLC separation of extracted compounds was performed with an ACQUITY UPLC BEH C18 column (2.1 × 100 mm, 1.7 μm particle size; Waters Corp.) at a flow rate of 0.35 mL/min. The mobile phases were composed as solvent A (0.1% formic acid in distilled water) and solvent B (acetonitrile), and analysis conditions were as follows: a gradient elution of 1% B at 0–1 min, 1–100% B at 1–7 min, 100% B at 7–8 min, 100–1% B at 8–8.2 min, 1% B at 8.2–10 min. 1 μL samples were injected in to a sampler. The conditions of electrospray ionization (ESI) were as follows: ramp collision energy, 10–30 V; capillary voltage, 2.5 kV; source temperature, 100 °C; desolvation temperature, 400 °C; cone voltage, 40 V; mass range, 50–1500 m/z. The data from the UPLC were analyzed using MarkerLynx software (Waters Corp.).

### 4.4. Cell Culture and Treatment

PC12 cells with the characteristics of adrenal gland blastoma cell lines were acquired from the Korean Cell Line Bank (Seoul, Korea), and incubated in MEM medium with 10% FBS, 50 units/mL penicillin, and 100 μg/mL streptomycin in the conditions of 5% CO_2_ at 37 °C.

### 4.5. Cell Viability

The in vitro neuronal cell viability was assessed by the MTT assay [52]. PC12 cells (10^4^ cells/well, *n* = 5) were treated with the sample. After 24 h, the cells were treated with H_2_O_2_, and incubated for 24 h. Finally, the cells were mixed with 5 mg/mL of MTT solution for 3 h, and the MTT was terminated by reacting DMSO. The contents of the MTT formazan were measured using a microplate reader (Epoch 2, BioTek Instruments, Inc., Winooski, VT, USA) at a test wavelength of 570 nm and reference wavelength of 690 nm.

### 4.6. Intracellular ROS

The intracellular ROS was measured using a DCF-DA [52]. PC12 cells (10^4^ cells/well, *n* = 5) were treated with the sample. After 24 h, the cells were treated with H_2_O_2_, and then were incubated for 3 h. The cells were treated with 10 μM DCF-DA dissolved in PBS. ROS production was measured using a fluorescence microplate reader (Infinite 200, Tecan Co., San Jose, CA, USA) at 485 nm excitation and with 530 nm emission filters.

### 4.7. Animals and Experimental Design

All procedures of the animal experiment followed the guidelines of the Animal Care and Use Committee of Gyeongsang National University (certificate: GNU-170605-M0023, 05 June 2017) and were performed in accordance with the Policy of the Ethical Committee of Ministry of Health and Welfare, Republic of Korea. Experiments were carried out using four-week-old male ICR mice purchased from Samtako (Osan, Korea). Experimental animals were divided into two or three per cage. These were housed in standard laboratory conditions with free access to fodder and water. The animals were orally fed a sample dissolved in drinking water through a stomach tube once a day for three weeks. When the diet was completed, Aβ1-42 was injected intracerebroventricularly (i.c.v.) as a single dose in the bregma (410 pM, 10 μL in saline) using a 25-μL Hamilton microsyringe combined with a 26-gauge needle that was inserted to a depth of 2.5 mm according to the ethical guideline. The Hamilton microsyringe needle was injected laterally at approximately 45° about 2 mm into the bregma of the skull [53]. The animals were randomly divided into groups: NC (saline—injection + water—oral administration, *n* = 13) group, Aβ (Aβ_1–42_—injection + water—oral administration, *n* = 13) group and GTO (Aβ_1–42_-injection + GTO-oral administration, *n* = 13) group (50 and 100 mg (with 2% Tween 80) /kg of body weight, *n* = 13, GTO 50 and GTO 100 groups, respectively). After Aβ injection, the experiment was divided into two groups. One group underwent cognitive in vivo testing (*n* = 8) and was sacrificed for ex vivo experiments (*n* = 8). The other group was immediately sacrificed for mitochondrial-related experiments after Aβ injection (*n* = 5).

### 4.8. Y-Maze Test

The Y-maze was made of white acrylic plate, and the length, height, and width of each arm were 33, 15, and 10 cm, respectively. The animals were located at the end of the designated arm, and these mice were allowed to move freely in the Y-maze for 8 min [54]. The movement of the mice and their paths were recorded using a video motion recognition system (Smart 3.0, Panlab, Barcelona, Spain).

### 4.9. Passive Avoidance Test

To estimate the ameliorating effect of short-term memory in Aβ-induced mice, a passive avoidance test was performed [55]. The test chamber was divided into a dark part that could give electrical stimulation and an illuminated part. First, the experimental animals were located in a light square, and 60 s later the entrance between the compartments was opened. When the four feet of the experimental animals had entered the dark cubicle, a foot shock could be applied (0.5 mA, 3 s), and the first latency time was recorded. In the test trial, the step-through latency for mice to re-enter the dark chamber was recorded after 24 h (maximum time: 300 s).

### 4.10. Morris Water Maze Test

A circular water pool (90 cm in diameter and 30 cm deep) was randomly separated into four zones, N, S, E, and W. In the center of the W quadrant, a white platform was submerged 1 cm below the water level. The experimental animals were allowed to swim freely, and the swimming activities were recorded using a SMART video tracking system (Smart 3.0, Panlab, Barcelona, Spain). Four training trials were conducted for each animal to swim and escape (repeated four times a day). Lastly, a probe test was conducted without the platform for 90 s, and the time they stayed in the W zone was determined [56].

### 4.11. Preparation of Brain Tissue

After the in vivo tasks, animals were fasted for 12 h, and then sacrificed using CO_2_ inhalation for ex vivo tests. The whole brain tissues removed were homogenized using a bullet blender (Next Advance Inc., Averill Park, NY, USA) with phosphate-buffered saline (PBS) for acetylcholine (ACh), acetylcholinesterase (AChE), malondialdehyde (MDA), and SOD assays, and with 10 mM phosphate buffer with 1 mM EDTA (pH 6.7) for a glutathione (GSH) assay at 4 °C. The cerebral protein concentration was measured with a Bradford protein assay [57].

### 4.12. Measurement of MDA

Homogenized mouse brain tissue extracted with PBS was mixed with 1% phosphoric acid and 0.67% thibarbituric acid. After reacting at 95 °C for 1 h, the mixture was centrifuged at 2500× *g* for 10 min. The absorbance was measured at 532 nm. The MDA content was expressed in micromoles per mg protein [52].

### 4.13. Measurement of SOD Contents

The tissue was centrifuged at 10,000× *g* using the same tissue as above. The supernatant was discarded and the pellet was taken. After 1× cell extraction buffer was added, mixing was performed every 5 min for 30 min. After centrifugation at 400× *g* for 10 min, the supernatant was used for the experiment. SOD content was determined with an SOD determination kit (Sigma-Aldrich Chemical Co.).

### 4.14. Measurement of Reduced GSH

The reduced GSH content in mouse brain was determined using the homogenized mouse brain with 10-fold phosphate buffer. The supernatant was obtained by centrifugation (10,000× *g*) for 15 min. The same amount of 5% metaphosphoric acid was added to the supernatant to remove the interference protein by spinning down at 2000 g. Once again, the supernatant was mixed with 0.26 M tris-HCl (pH 7.8), 0.65 N NaOH, and 1 mg/mL OPT, and reacted at room temperature while light was blocked for 15 min. Fluorescence was measured at a wavelength of 320 nm (excitation filter) and 420 nm (emission filter) using a fluorometer (Infinite F200, Tecan Co., San Jose, CA, USA). The reduced GSH content was assigned to a standard curve [58].

### 4.15. Measurement of ACh

The homogenized brain tissue was centrifuged at 14,000× *g*. This supernatant, with alkaline hydroxylamine reagent [3.5 N sodium hydroxide and 2 M hydroxylamine in HCl] added, was reacted for 1 min at room temperature, and then 0.5 N HCl (pH 1.2) and 0.37 M FeCl_3_ in 0.1 N HCl were reacted. The absorbance was measured at 540 nm [59].

### 4.16. Measurement of AChE Activity

The same supernatant as above was used for enzyme experiments. This supernatant and 50 mM sodium phosphate buffer (pH 8.0) were pre-incubated at 37 °C for 15 min. After adding an Ellman’s reaction mixture, the absorbance was measured at 405 nm. AChE activity was expressed as a relative value compared to the activity of the NC group [60].

### 4.17. Mitochondrial Extraction in Brain Tissue

Mouse brain tissue was homogenized in a mitochondria isolation (MI) buffer [215 mM mannitol, 75 mM sucrose, 0.1% BSA, 20 mM HEPES sodium salt (pH 7.2)] containing 10 mM of 1 mM EGTA. The homogenate was spun down at 1300× *g* for 10 min at 4 °C. The supernatant was centrifuged again at 13,000× *g* for 10 min at 4 °C. To remove the synaptosome, the supernatant was removed and the remaining mitochondrial pellet was mixed with MI buffer containing 0.1% digitonin. After 5 min of reaction, the mixture was added to 2 mL of MI buffer containing 1 mM EGTA and centrifuged at 13,000× *g* for 15 min at 4 °C. The pellet was mixed in MI buffer and centrifuged again at 10,000× *g* for 10 min. The pellet was added to the MI buffer to conduct the experiment.

### 4.18. Measurement of Mitochondrial ROS

To investigate the amount of ROS in mitochondria, the isolated mitochondria were incubated with KCl-based respiration buffer [125 mM potassium chloride, 2 mM potassium phosphate monobasic, 20 mM HEPES, 1 mM magnesium chloride, 500 μM EGTA, 2.5 mM malate and 5 mM pyruvate] and DCF-DA for 20 min. After incubation, fluorescence was measured [61].

### 4.19. Measurement of MMP

To assess the mitochondrial membrane potential (MMP), the isolated mitochondria were mixed with 1 mM JC-1 in MI buffer containing 5 mM pyrivate and 5 mM malate. This mixture was reacted at room temperature for 20 min in a dark room, and fluorescence (excitation: 530/25 nm, emission: 590/35 nm) was measured [61].

### 4.20. Western Blotting

Western blot analysis was performed by homogenizing the brain of the experimental animals (*n* = 3) with lysis buffer containing 50 mM Tris-HCl (pH 7.4), 150 mM sodium chloride, 1 mM ethylenediaminetetraacetic acid, 0.25% sodium deoxycholate, 1% NP40, 1 mM Na_3_VO_4_, 1 mM PMSF and 1% protease inhibitors. These homogenates were centrifuged at 13,000× *g*, 4 °C for 10 min. After the protein content was adjusted to the same concentration, proteins were electrophoresed on SDS-PAGE, and transferred to a polyvinylidene difluoride (PVDF) membrane (Millipore, Billerica, MA, USA). To prevent the nonspecific binding of other proteins, the membranes were blocked with 5% skim milk. After incubation at 4 °C for 12 h with A primary antibody (1:1000) containing AChE, tumor necrosis factor-alpha (TNF-α), phosphorylated c-Jun N-terminal kinase (p-JNK), phosphorylated Akt (p-Akt), phosphorylated tau (p-tau), phosphorylated nuclear factor kappa-light-chain-enhancer of activated B cells (p-NF-κB), Bcl-2-associated X protein (BAX), cytosolic cytochrome c, mitochondrial cytochrome c, and β-actin, the secondary antibody (1:2000) was incubated for 1 h. The Western blot image was detected using a Western blot imager (iBright Imager, Thermo Fisher Scientific, Waltham, MA, USA), and the expression densities were calculated using image detecting software (Image J software, National Institutes of Health, Bethesda, MD, USA).

### 4.21. Statistical Analysis

All data were expressed as the mean ± standard deviation (SD). The statistical significance was analyzed by a one-way analysis of variance (ANOVA). The significance of differences was conducted by using Duncan’s new multiple-range test (*p* < 0.05) of SAS ver. 9.4 (SAS Institute Inc., Cary, NC, USA).

## 5. Conclusions

In summary, these results suggest that GTO has a significant neuroprotective effect against H_2_O_2_-induced cytotoxicity in PC12 cells by increasing cell viability and reducing ROS contents. Also, GTO ameliorated cognitive and behavioral dysfunction in Aβ_1–42_-injected ICR mice. GTO protected the antioxidant system and cholinergic function and inhibited mitochondrial dysfunction via the Aβ-related JNK/Akt pathway and Akt/apoptosis pathway. In this study, it is suggested that GTO, a byproduct of green tea seed, could be used as a functional food material with the potential for preventing cognitive dysfunction and nerve damage. Through this study, not only can the industrial value be increased by using green tea oil, which has previously been discarded as a byproduct, but it could also be used as a nutraceutical material by examining the protective effect on neuronal cells and the ameliorating effect on cognitive function. In addition, it is considered that additional studies on various physiological activities should be conducted to confirm the physiologically active material of GTO as a nutraceutical. Lastly, the correlation between the individual structural characteristics and the functionality of the identified physiologically active substances should be further investigated.

## Figures and Tables

**Figure 1 ijms-20-01865-f001:**
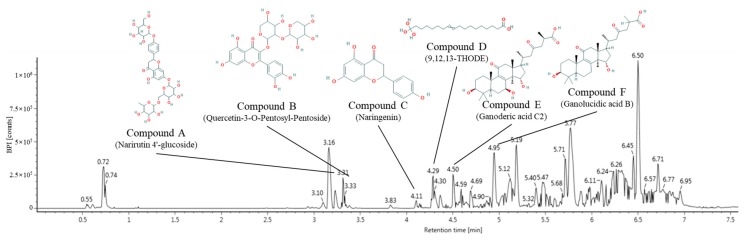
Q-TOF UPLC/MS chromatographic spectra in negative ion mode.

**Figure 2 ijms-20-01865-f002:**
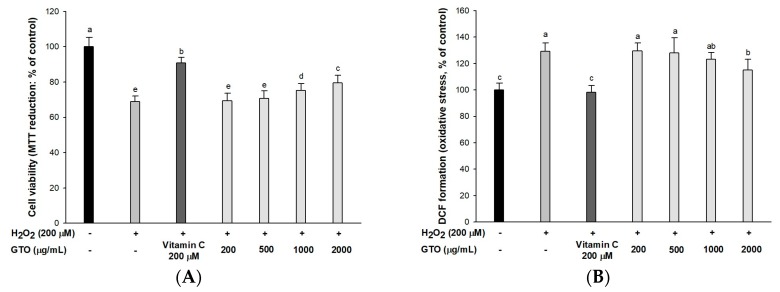
Neuroprotective effect of green tea seed oil (GOT) in PC12 cells. (**A**) Cell viability; (**B**) cellular oxidative stress. Results shown are mean ± SD (*n* = 3). Data were statistically represented at *p* < 0.05, and different lowercase letters indicate statistical significance.

**Figure 3 ijms-20-01865-f003:**
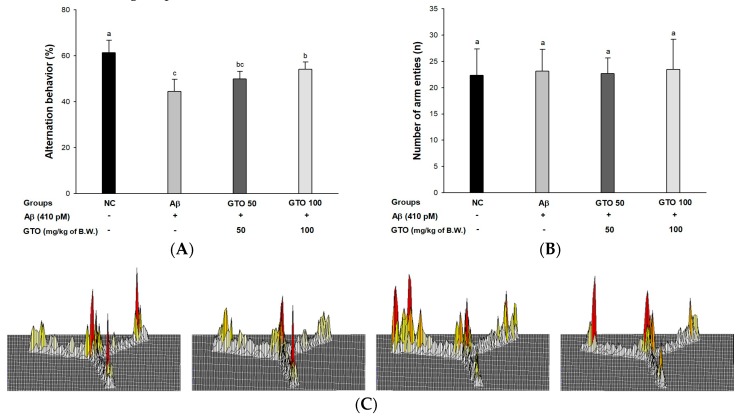
Protective effect of green tea seed oil (GOT) on Y-maze test in Aβ-induced mice. (**A**) Spontaneous alternation behavior; (**B**) number of arm entries; (**C**) 3D moving routes. Results shown are mean ± SD (*n* = 8). Data were statistically represented at *p* < 0.05, and different lowercase letters indicate statistical significance. NC: Aβ^−^/sample^−^ group; Aβ: Aβ^+^/sample^−^ group; GTO 50: Aβ^+^/GTO (50 mg/kg of body weight)^+^ group; GTO 100: Aβ^+^/GTO (100 mg/kg of body weight)^+^ group. B.W.: body weight.

**Figure 4 ijms-20-01865-f004:**
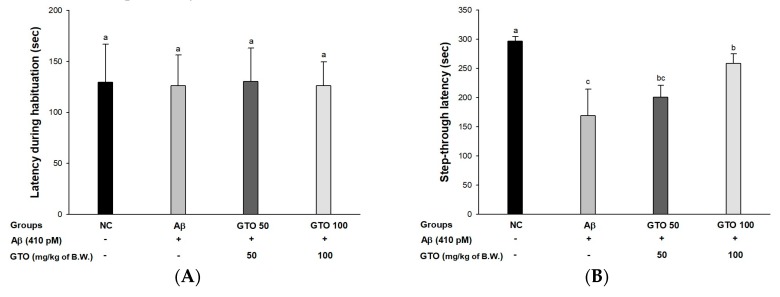
Protective effect of green tea seed oil (GOT) on passive avoidance test in Aβ-induced mice. (**A**) Latency during habituation; (**B**) step-through latency. Results shown are mean ± SD (*n* = 8). Data were statistically represented at *p* < 0.05, and different lowercase letters indicate statistical significance. NC: Aβ^−^/sample^−^ group; Aβ: Aβ^+^/sample^−^ group; GTO 50: Aβ^+^/GTO (50 mg/kg of body weight)^+^ group; GTO 100: Aβ^+^/GTO (100 mg/kg of body weight)^+^ group. B.W.: body weight.

**Figure 5 ijms-20-01865-f005:**
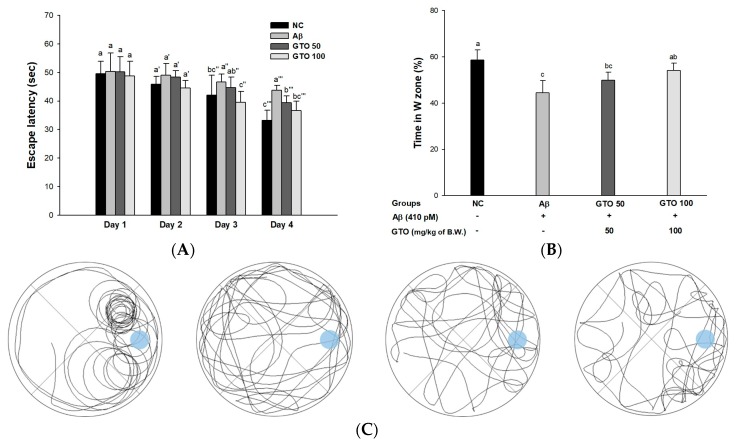
Protective effect of green tea seed oil (GOT) on Morris water maze test in Aβ-induced mice. (**A**) Escape latency in the training trial; (**B**) retention time on W zone in the probe trial (**C**) path tracing of each groups in the probe trial. Results shown are mean ± SD (*n* = 8). Data were statistically represented at *p* < 0.05, and different lowercase letters indicate statistical significance. NC: Aβ^−^/sample^−^ group; Aβ: Aβ^+^/sample^−^ group; GTO 50: Aβ^+^/GTO (50 mg/kg of body weight)^+^ group; GTO 100: Aβ^+^/GTO (100 mg/kg of body weight)^+^ group. B.W.: body weight.

**Figure 6 ijms-20-01865-f006:**
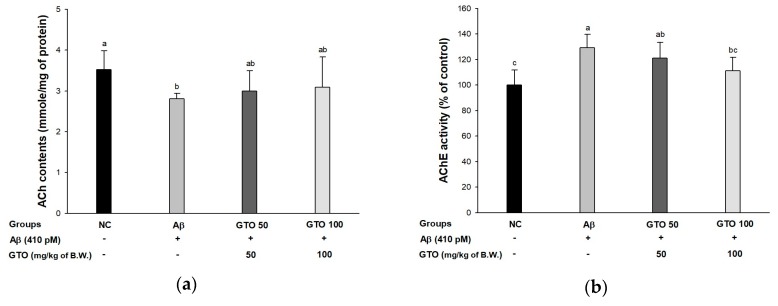
Protective effect of green tea seed oil (GOT) on Aβ-induced cholinergic dysfunction. (**a**) ACh levels; (**b**) AChE activities; (**c**) protein expression levels; (**d**) representative Western blots for total protein and expression of AChE in mice brain tissues. Results shown are mean ± SD (*n* = 8). Data were statistically represented at *p* < 0.05, and different lowercase letters indicate statistical significance. NC: Aβ^−^/sample^−^ group; Aβ: Aβ^+^/sample^−^ group; GTO 50: Aβ^+^/GTO (50 mg/kg of body weight)^+^ group; GTO 100: Aβ^+^/GTO (100 mg/kg of body weight)^+^ group. B.W.: body weight.

**Figure 7 ijms-20-01865-f007:**
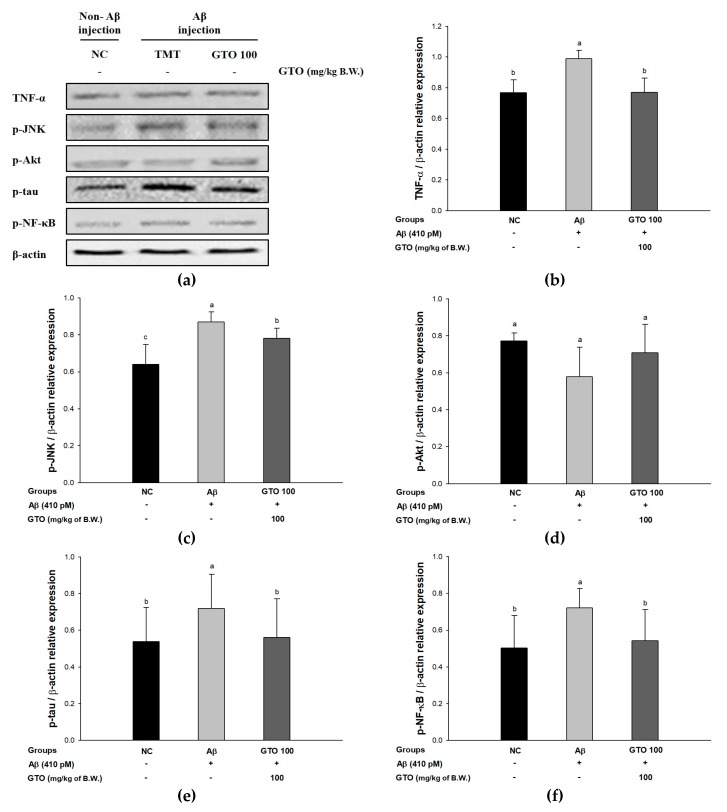
Ameliorating effect of green tea seed oil (GOT) on Aβ-related JNK/Akt pathway in mice brain tissues. (**a**) Representative Western blots for total protein and expression of TNF-α, p-JNK, p-Akt, p-tau, p-NF-κB and β-actin. The protein expression levels of TNF-α (**b**); p-JNK (**c**); p-Akt (**d**); p-tau (**e**) and p-NF-κB (**f**). Results shown are mean ± SD (*n* = 3). Data were statistically represented at *p* < 0.05, and different lowercase letters indicate statistical significance. NC: Aβ^−^/sample^−^ group; Aβ: Aβ^+^/sample^−^ group; GTO 100: Aβ^+^/GTO (100 mg/kg of body weight)^+^ group. B.W.: body weight.

**Figure 8 ijms-20-01865-f008:**
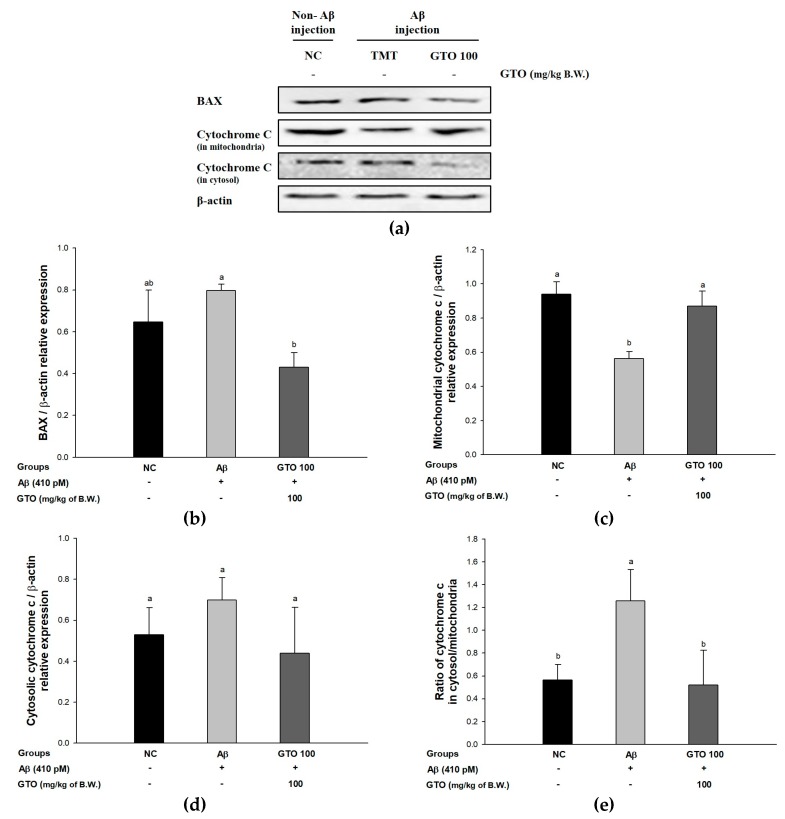
Ameliorating effect of green tea seed oil (GOT) on Aβ-related Akt/apoptosis pathway in mice brain tissues. (**a**) Representative Western blots for total protein and expression of BAX, mitochondrial cytochrome c, cytosolic cytochrome c and β-actin; The protein expression levels of BAX (**b**); mitochondrial cytochrome c (**c**); cytosolic cytochrome c (**d**); mitochondrial cytochrome c/cytosolic cytochrome c ratio (**e**) Results shown are mean ± SD (*n* = 3). Data were statistically represented at *p* < 0.05, and different lowercase letters indicate statistical significance. NC: Aβ^−^/sample^−^ group; Aβ: Aβ^+^/sample^−^ group; GTO 100: Aβ^+^/GTO (100 mg/kg of body weight)^+^ group. B.W.: body weight.

**Table 1 ijms-20-01865-t001:** Compounds identified from green tea seed oil (GTO).

No.	RT ^a^ (min)	Parent Ion ^b^ (m/z)	MS^2^ Ions ^c^ (*m*/*z*)	Compound
**1**	3.31	741	271, 459, 579, 595, **621**	Narirutin 4′-glucoside
**2**	3.33	565	**271**, 301	Quercetin-3-O-pentosyl-pentoside
**3**	4.11	271	199, **151**, 177	Naringenin
**4**	4.29	329	139, 171, 183, **211**, 229	9,12,13-THODE
**5**	4.50	517	**455**, 499	Ganoderic acid C2
**6**	4.95	501	393, 421, 423, **439**, 483	Ganolucidic acid B

^a^ RT means retention time. ^b^ Ions are presented at *m*/*z* [M–H]^−^. ^c^ Bold indicates the main fragment of MS^2^.

**Table 2 ijms-20-01865-t002:** Protective effect of green tea seed oil (GOT) on Aβ-induced biochemical changes related to the antioxidant system.

Parameters	Groups			
	NC	Aβ	GTO 50	GTO 100
MDA	40.88 ± 2.97 ^c^	54.52 ± 5.03 ^a^	47.62 ± 4.80 ^b^	44.93 ± 5.84 ^bc^
SOD	44.18 ± 3.96 ^a^	30.73 ± 6.30 ^b^	35.99 ± 10.88 ^ab^	38.85 ± 4.23 ^ab^
reduced GSH	2.79 ± 0.33 ^ab^	2.02 ± 0.29 ^b^	2.14 ± 0.49 ^b^	3.22 ± 0.45 ^a^

Results shown are mean ± SD (*n* = 8). Data were statistically represented at *p* < 0.05, and different lowercase letters indicate statistical significance. NC: Aβ^−^/sample^−^ group; Aβ: Aβ^+^/sample^−^ group; GTO 50: Aβ^+^/GTO (50 mg/kg of body weight)^+^ group; GTO 100: Aβ^+^/GTO (100 mg/kg of body weight)^+^ group. B.W.: body weight.

**Table 3 ijms-20-01865-t003:** Protective effect of green tea seed oil (GOT) on Aβ-induced mitochondrial dysfunction.

Parameters	Groups			
	NC	Aβ	GTO 50	GTO 100
ROS	100.00 ± 19.83 ^b^	140.41 ± 20.24 ^a^	130.52 ± 2.88 ^ab^	106.47 ± 13.36 ^ab^
MMP	100.00 ± 5.12 ^a^	68.74 ± 6.17 ^b^	79.02 ± 7.31 ^ab^	80.04 ± 16.16 ^ab^

Results shown are mean ± SD (*n* = 8). Data were statistically represented at *p* < 0.05, and different lowercase letters indicate statistical significance. NC: Aβ^−^/sample^−^ group; Aβ: Aβ^+^/sample^−^ group; GTO 50: Aβ^+^/GTO (50 mg/kg of body weight)^+^ group; GTO 100: Aβ^+^/GTO (100 mg/kg of body weight)^+^ group. B.W.: body weight.

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
