# Peer review of "Green Tea Seed Oil Suppressed Aβ1–42-Induced Behavioral and Cognitive Deficit via the Aβ-Related Akt Pathway"

_ijms, 2019, doi:10.3390/ijms20081865_

Round 1
Reviewer 1 Report
The subject is interesting and well described. Some suggestions as follows:
The Introduction should be improved: the authors should insert a general discussion on importance of nutraceuticals and antioxidants and a better description of the main components of green tea.
In this regards appropriate reference should be added:
Santini A, Novellino E. To Nutraceuticals and Back: Rethinking a Concept. Foods. 2017 5,6(9). pii: E74. doi: 10.3390/foods6090074.
Santini, A.; Tenore, G.C.; Novellino, E. Nutraceuticals: A paradigm of proactive medicine. European J. Pharmac.l Sci., 2017, 96, 53-61.
Daliu, P.; Santini, A.; Novellino, E. From pharmaceuticals to nutraceuticals: bridging disease prevention and management. Expert Review of Clinical Pharmacology, 2018, 28, 1-7.
Durazzo, A.; D'Addezio, L.; Camilli, E.; Piccinelli, R.; Turrini, A.; Marletta, L.; Marconi, S.; Lucarini M.; Lisciani, S.; Gabrielli, P.; Gambelli ,L.; Aguzzi, A.; Sette, S. From Plant Compounds to Botanicals and Back: A Current Snapshot. Molecules. 2018 Jul 24;23(8). pii: E1844. doi: 10.3390/molecules23081844.
Durazzo, A. Extractable and Non-extractable polyphenols: an overview. In: Saura-Calixto, F.; Pérez-Jiménez, J. (Eds). Non-extractable Polyphenols and Carotenoids: Importance in Human Nutrition and Health. Food Chem., Func. Anal. No. 5, Royal Society of Chemistry, London, UK, 2018 , pp. 37.
Zhang M, Zhang X, Ho CT, Huang Q. Chemistry and Health Effect of Tea Polyphenol (-)-Epigallocatechin 3- O-(3- O-Methyl)gallate.J Agric Food Chem. 2018 Oct 22. doi: 10.1021/acs.jafc.8b04837.
Grzesik M, Naparło K, Bartosz G, Sadowska-Bartosz I. Antioxidant properties of catechin: comparison with other antioxidants.Food Chem. 2018 Feb 15;241:480-492. doi: 10.1016/j.foodchem.2017.08.117.
In the conclusion the authors should add a final paragraph on the advantage and limits of research.
Author Response
A List of Changes
Manuscript ID: ijms-301505
Corresponding Author: Ho Jin Heo
Division of Applied Life Science,
Institute of Agriculture and Life Science,
Gyeongsang National University, Jinju 52828, Korea
TEL: 82-55-772-1907
FAX: 82-55-772-1909
E-mail: [email protected]
Dear Editor (Owen Chen):
We followed most part of the reviewers’ suggestions, and revised accordingly. In addition, we responded some questions in detail to explain our points of view based on reported facts. Once again, thank you for your time and consideration. I hope that the revised manuscript will be acceptable to you, and thank you for the valuable suggestions.
Thank you!
Best wishes,
Ho Jin Heo
[The replies on the review of ‘Reviewer #1’]
Comments:
Ø Dear Authors,
Manuscript IJMS-301505 deals with a research topic of interest. However, revision in English language is mandatory prior publication. My specific comments follow the text sequence:
→ Thank you for your sincere suggestion. Depending on your opinion, we have reviewed all of your concerns. The part of the review was marked in red in the text.
(Line 93). Change ''Also'' to ''Reversely,''
Also, intracellular ROS in the H2O2-treated ~
→ Reversely, intracellular ROS in the H2O2-treated ~
(Line 105 and within the whole text). Change ''compared with '' to ''compared to''.
→ We changed the whole text "compared with" to "compared to".
(Line 107). Change ''But'' to ''On the other hand''.
But, that of the EFDK groups was similar to the NC group (Figure 2C).
→ On the other hands, that of the EFDK groups was similar to the NC group (Figure 2C).
(Line 129). Change ''increased'' to ''increasing''.
However, the EFDK-treated groups (27.78% and 29.82%, respectively) showed increased time in the W zone.
→ However, the EFDK-treated groups (27.78% and 29.82%, respectively) showed increasing time in the W zone. (Figure 2C).
(Line 144). Change ''But'' to ''However''
But that of the EFDK groups (3.20 nmole/mg of protein and 3.04 nmole/mg of protein) were reduced compared to the TMT group.
→ However, that of the EFDK groups (3.20 nmole/mg of protein and 3.04 nmole/mg of protein) were reduced compared to the TMT group.
(Line 155).''In order to estimate, ....''.
To estimate the protective effect of the cholinergic system, acetylcholinesterase (AChE) activity and acetylcholine (ACh) content were examined (Figure 6).
→ In order to estimate the protective effect of the cholinergic system, acetylcholinesterase (AChE) activity and acetylcholine (ACh) content were examined (Figure 6).
(Line 193). Change ''Also'' to ''In addition,''.
Also, the expression of p-Akt of the TMT group (15.61%) decreased more than for the NC group.
→ In addition, the expression of p-Akt of the TMT group (15.61%) decreased more than for the NC group.
(Line 203).''In order to, ...''.
To investigate the regulating effect of the apoptotic signaling pathway, western blotting was carried out (Figure 9).
→ In order to investigate the regulating effect of the apoptotic signaling pathway, western blotting was carried out (Figure 9).
(Line 205).''was increased compared to..''.
The expression of Bcl-2-associated X protein (BAX) and cytosolic cytochrome c of the TMT group (29.31% and 88.85%, respectively) was increased compared with the NC group.
→ The expression of Bcl-2-associated X protein (BAX) and cytosolic cytochrome c of the TMT group (29.31% and 88.85%, respectively) was increased compared to the NC group.
(Line 244).''Oxidative stress, caused by oxygen,...''.
Oxidative stress, with oxygen, is the result of the metabolism of aerobic cells in the body.
→ Oxidative stress, caused by oxygen, is the result of the metabolism of aerobic cells in the body.
(Line 248).'' Consecutive oxidative stress can cause....''.
Increased stress from this reaction can be reduced with the consumption of antioxidants such as vitamins and various flavonoids.
→ Consecutive oxidative stress can cause with the consumption of antioxidants such as vitamins and various flavonoids.
(Line 265). ''In the present study it was...''.
Indirectly, we confirmed that EFDK has a protective effect on hippocampal neurons with an in vitro analysis of HT22 cells (Figure 1).
→ In the present study, it was confirmed that EFDK has a protective effect on hippocampal neurons with an in vitro analysis of HT22 cells (Figure 1).
(Line 299). ''Furthermore, it was reported....''.
EFDK contains various flavonoids.
→ Furthermore, it was reported that EFDK contains various flavonoids.
(Line 563). ''Furthermore, ...''.
It also ameliorated mitochondrial function by regulating the expression of protein signaling via JNK/Akt and apoptotic pathways (Figure 11).
→ Furthermore, it also ameliorated mitochondrial function by regulating the expression of protein signaling via JNK/Akt and apoptotic pathways (Figure 11).
Based on the aforementioned,
I suggest a minor revision prior publication.
Thanks for valuable suggestions!
We conclude the answer for your general review and detailed requirements above.
Also, if you have any additional supplements after checking, sincerely hope to modify.
Thank you for your attention.

Reviewer 2 Report
The manuscript entitled “Green tea seed oil suppressed Aβ1-42-induced behavioral and cognitive deficit via Aβ-related Akt pathway ” presents interesting issue, but it requires some important corrections.
General:
Authors should avoid personal forms (e.g. our study) and they should use rather not personal ones (e.g. the study).
Abstract:
Authors should briefly (1-2 sentences) justify the study (indicate why it was needed)
Authors should present applied methodology.
Authors should present specific results.
Introduction:
Authors should present the detailed characteristics of the green tea (components, amount of them, their possible actions, applications, etc.)
Authors should briefly formulate the aim of their study (e.g. “The aim of the study was…”).
Results:
Figure 2, 3a, 3b, 4, 5a, 5b, 6, 7a, 7b, 7d, 8, 9b-f, 10b-e – Authors should rather present their data as tables to be easier to follow
It seems, that Authors did not verify the normality of distribution for the assessed variables. Authors must verify the normality of distribution and specify the test applied for verification.
If the distribution is normal, the mean values should be presented (accompanied by SD), but if it is different than normal, the median, accompanied by minimum and maximum values should be presented – it should be specified that distribution is normal if it is.
The applied test should be chosen taking into account the observed distribution.
Discussion:
Authors should in this section directly discuss the obtained results, instead of reproducing basic information from Introduction section, or presenting basic information that should be rather mentioned in the Introduction section. This section should be associated with the conducted study and the obtained results.
In this section, Authors should avoid referring figures presented in the previous parts of the study.
Authors should extensively discuss the limitations of the study.
Materials and Methods:
The number of repetitions should be presented for each analysis.
It seems, that Authors did not verify the normality of distribution for the assessed variables. Authors must verify the normality of distribution and specify the test applied for verification.
If the distribution is normal, the mean values should be presented (accompanied by SD), but if it is different than normal, the median, accompanied by minimum and maximum values should be presented – it should be specified that distribution is normal if it is.
The applied test should be chosen taking into account the observed distribution.
Conclusions:
Figure should not be presented in this section.
Author Contributions:
* is not needed to be indicated in this section.
Author Response
A List of Changes
Manuscript ID: ijms-301505
Corresponding Author: Ho Jin Heo
Division of Applied Life Science,
Institute of Agriculture and Life Science,
Gyeongsang National University, Jinju 52828, Korea
TEL: 82-55-772-1907
FAX: 82-55-772-1909
E-mail: [email protected]
Dear Editor (Owen Chen):
We followed most part of the reviewers’ suggestions, and revised accordingly. In addition, we responded some questions in detail to explain our points of view based on reported facts. Once again, thank you for your time and consideration. I hope that the revised manuscript will be acceptable to you, and thank you for the valuable suggestions.
Thank you!
Best wishes,
Ho Jin Heo
[The replies on the review of ‘Reviewer #2’]
Comments:
Ø The manuscript presents a very good research on a very well known plant with a great interest for the journal readers.
Still, the article can be improved by small modifications.
1. Trimethyltin is infact trimethyltin chloride
→ Thank you for your sincere suggestion. Depending on your opinion, the first mentioned trimethyltin (TMT) was labeled as trimethyltin chloride (TMT).
(Lines 16-18)
This study was conducted to assess the antioxidant capacity and protective effect of ethyl acetate fraction from persimmon (Diospyros kaki) (EFDK) on H2O2-induced hippocampal HT22 cells and trimethyltin (TMT)-induced ICR mice.
→ This study was conducted to assess the antioxidant capacity and protective effect of ethyl acetate fraction from persimmon (Diospyros kaki) (EFDK) on H2O2-induced hippocampal HT22 cells and trimethyltin chloride (TMT)-induced ICR mice.
(Lines 62-64)
Thus, this study aimed to investigate the ameliorating effect of ethyl acetate fraction from persimmon (EFDK) in trimethyltin (TMT)-induced ICR mice by conducting an in vivo test, biochemical assay and signaling analysis related to oxidative stress and AD pathology.
→ Thus, this study aimed to investigate the ameliorating effect of ethyl acetate fraction from persimmon (EFDK) in trimethyltin chloride (TMT)-induced ICR mice by conducting an in vivo test, biochemical assay and signaling analysis related to oxidative stress and AD pathology.
2. Section 4.1. Some details are missing and it makes quite difficult to reproduce the extract. This section should be improved for future researchers who try to prepare the fractions. Like what was the ration of plant produce to ethanol 80%? The sample was suspended in 300 ml solvent, but what of the quantity of product suspended?
→ Thank you for your feedback. Based on your opinion, we added the sample extraction process in more detail.
(Lines 383-388)
Sample extraction was performed with 50-fold 80% ethanol at 40°C for 2 h. The extracted sample was filtered and concentrated using a vacuum rotary evaporator (N-N series, Eyela Co., Tokyo, Japan). This sample was re-suspended in 300 mL of distilled water, and successively fractionated with 300 mL of n-hexane, chloroform and ethyl acetate.
→ Sample (60 g) was extracted with 50-fold 80% ethanol at 40°C for 2 h. The extracted sample was filtered and concentrated using a vacuum rotary evaporator (N-N series, Eyela Co., Tokyo, Japan). The yield of the 80% ethanol extract was 36.34% of dried weight. This extracted sample (21.80 g) was re-suspended in 300 mL of distilled water, and successively fractionated with 300 mL of n-hexane, chloroform and ethyl acetate.
3. A fragment of the 4.4 section and 4.7 section about the mice overlap. It should be corrected not to be repeated.
→ Thank you for your sincere suggestion. There was a small mistake in the text. The duplicate content was deleted, and the text was modified.
(Lines 411-414)
To measure the inhibitory effect of lipid peroxide using mouse brain tissue, ICR (male, 4 weeks old) was purchased from a laboratory animal supplier (Samtako, Osan, Korea). Experimental animals were kept in constant temperature (22 ± 2°C) and constant humidity (50-55%). All animal experiments received the approval of the Animal Care and Use Committee of Gyeongsang National University (certificate: GNU-131105-M0067), and were carried out according to the provisions of the Policy of the Ethical Committee of the Ministry of Health and Welfare, Republic of Korea. Brain tissue was harvested and homogenized by adding 20 mM Tris-HCl buffer (pH 7.4), and centrifuged at 12,000 g, for 15 min at 4°C.
→ To measure the inhibitory effect of lipid peroxide using mouse brain tissue, ICR (male, 4 weeks old) was purchased from a laboratory animal supplier (Samtako, Osan, Korea). Experimental animals were kept in constant temperature (22 ± 2°C) and constant humidity (50-55%). All animal experiments received the approval of the Animal Care and Use Committee of Gyeongsang National University (certificate: GNU-131105-M0067), and were carried out according to the provisions of the Policy of the Ethical Committee of the Ministry of Health and Welfare, Republic of Korea. Brain tissue was harvested and homogenized by adding 20 mM Tris-HCl buffer (pH 7.4), and centrifuged at 12,000 g, for 15 min at 4°C.
4. How much TMT was used? How was the extract been administred? What vehicle was it used?
→ Thank you for your sincere suggestion. Based on your advice, we added the following.
(Lines 436-445)
The male ICR mice (4 weeks old) were purchased from an animal supplier (Samtako, Osan, Korea). For these experiments, the animals were divided to four groups: NC group (vehicle-intraperitoneally (i.p.) injected/vehicle-administration), TMT group (TMT-injected/ vehicle oral administration), and EFDK 10 and EFDK 20 groups (TMT-injected/EFDK 10 and 20 mg/kg of body weight oral administration, respectively). All animal experiments received the approval of the Animal Care and Use Committee of Gyeongsang National University (certificate: GNU-170605-M0023), and were carried out according to the provisions of the Policy of the Ethical Committee of the Ministry of Health and Welfare, Republic of Korea. The experiment design is presented in Figure 10.
→ The male ICR mice (4 weeks old) were purchased from an animal supplier (Samtako, Osan, Korea). For these experiments, the animals were divided to four groups: NC group (vehicle-intraperitoneally (i.p.) injected/vehicle-administration), TMT group (TMT-injected/ vehicle oral administration), and EFDK 10 and EFDK 20 groups (TMT-injected/EFDK 10 and 20 mg/kg of body weight oral administration, respectively). The vehicle was used as 0.85% sodium chloride solution, and the injected TMT concentration was 7.1 μg / kg of body weight. Samples were taken directly into the stomach using the stomach tube. All animal experiments received the approval of the Animal Care and Use Committee of Gyeongsang National University (certificate: GNU-170605-M0023), and were carried out according to the provisions of the Policy of the Ethical Committee of the Ministry of Health and Welfare, Republic of Korea. The experiment design is presented in Figure 10.
5. In my opinion, the discussion section is way too long. There is so much information, it becomes difficult to follow. It should be shorten and made simpler.
→ Thank you for your opinion. Based on your advice, we revised the manuscript.
(Lines 225-226)
This Aβ aggregation leads to neuronal death and ultimately leads to cognitive dysfunction and AD. In addition, the injured tissue itself induces the production of ROS, which may lead to a series of nerve cell deaths [3].
→ This Aβ aggregation leads to neuronal death and ultimately leads to cognitive dysfunction and AD. In addition, the injured tissue itself induces the production of ROS, which may lead to a series of nerve cell deaths [3].
(Lines 245-246)
It is essential for the growth of all cells, regardless of the type of cell, but it can be dangerous if it is produced in excess [3]. About 1-2% of the oxygen normally consumed in the body is converted to ROS, but cells with abnormalities in the antioxidant system have an increased proportion of ROS [4].
→ It is essential for the growth of all cells, regardless of the type of cell, but it can be dangerous if it is produced in excess [3]. About 1-2% of the oxygen normally consumed in the body is converted to ROS, but cells with abnormalities in the antioxidant system have an increased proportion of ROS [4].
(Lines 269-272)
TMT shows loss of the hippocampal CA3 region of the mouse, indicating loss of memory impairment and pyramidal neurons, and inhibits the connection of synaptic circuits from CA3 to CA1 [12]. Thus, TMT-induced neuronal damage is brought about by the induction of oxidative stress, which results in cognitive dysfunction similar to AD by inhibiting the connections between hippocampal regions.
→ TMT shows loss of the hippocampal CA3 region of the mouse, indicating loss of memory impairment and pyramidal neurons, and inhibits the connection of synaptic circuits from CA3 to CA1 [12]. Thus, TMT-induced neuronal damage is brought about by the induction of oxidative stress, which results in cognitive dysfunction similar to AD by inhibiting the connections between hippocampal regions [12].
(Lines 285-290)
In many studies, the brain has been identified as abnormally sensitive to external stress, and it is known that it is also easy for peroxidation of the brain membranes to occur [27]. The brain accounts for 2% of the body’s content, but consumes 20% of the total oxygen, when compared to other tissues. It contains fatty acids that are more sensitive to lipid peroxidation, and also not rich in antioxidants (about 10% compared to liver) [28]. Because of these conditions, brain tissue is considered to be sensitive to oxidative stress [27]. Oxidative stress can be eradicated by antioxidant substances. It has been reported that antioxidants can neutralize free radicals and inhibit the death of neurons [29]. Oxidative damage is known to increase the inflammatory response and lower levels of antioxidant enzymes in AD brain.
→ In many studies, the brain has been identified as abnormally sensitive to external stress, and it is known that it is also easy for peroxidation of the brain membranes to occur [27]. The brain accounts for 2% of the body’s content, but consumes 20% of the total oxygen, when compared to other tissues. It Brain tissue contains fatty acids that are more sensitive to lipid peroxidation, and also not rich in antioxidants (about 10% compared to liver) [28]. Because of these conditions, brain tissue is considered to be sensitive to oxidative stress [27]. Oxidative stress can be eradicated by antioxidant substances. It has been reported that antioxidants can neutralize free radicals and inhibit the death of neurons [29]. Oxidative damage is known to, and increase the inflammatory response and lower levels of antioxidant enzymes in AD brain [29]
6. Also, the style should be more realistic. The effect of the extract are intersting, but they are not that impressive as the authors lead us to belive. For example, ABTS effect is 5 time less than ascorbic acid, DPPH more than 10 times less compared to ascorbic acid. Also, the model of TMT is far from perfect. If the extract produced the presented effects on TMT mice, it does not mean it will do the same in AD or Parkinson mice! In my opinion, the extract ameliorated the effect of TMT and all the discussion section are assumptions.
→ Thank you for your sincere suggestion. As you mentioned, EFDK showed lower antioxidant activity 5 times that of ABTS and 10 times of DPPH when compared with vitamin c. Vitamin C is a very powerful antioxidant. However, when a large amount of vitamin c is ingested, it is released into the urine without accumulating more than a certain amount in the body. Also, in our previous study, the group receiving the same amount of vitamin c showed neurotoxicity. In general, it can be assumed that a material with excellent antioxidant properties can help cognitive function. However, in spite of low antioxidant activity compared to vitamin c, EFDK showed excellent cognitive function improvement by in vivo and ex vivo tests.
→ Also, TMT model is used as AD model in various articles. TMT is an organotin compound with potent neurotoxicity and is known to cause significant damage to the hippocampus, hypothalamus and amygdala in the brains. This TMT-induced deficit causes behavioral and learning dysfunction. It induces oxidative stress and promotes the death of nerve cells. When exposed to neuronal cells, oxidative stress produced by TMT promotes calcium overload and mitochondrial damage. And it leads to increase in contents of reactive oxygen species (ROS) and reactive nitrogen species (RNS), and increases lipoperoxidation. In addition, TMT may cause damage to the cholinergic system and inhibit neurotransmission as described in the text. This mechanism is similar to the pathogenesis of AD. For this reason, the TMT model is similar to the AD model, and may play an important role in studying oxidative stress and damage to the cholinergic system. We have confirmed the ameliorating effects of EFDK on oxidative stress and cholinergic system damage in this study. Thus, we will carry out the improvement of EFDK against neurotoxicity and neuro-inflammation using amyloid beta1-42-induced mouse model.
(Lines 563-567)
Furthermore, it also ameliorated mitochondrial function by regulating the expression of protein signaling via JNK/Akt and apoptotic pathways (Figure 11).
→ Furthermore, it also ameliorated mitochondrial function by regulating the expression of protein signaling via JNK/Akt and apoptotic pathways (Figure 11).
We have confirmed the ameliorating effects of EFDK on oxidative stress and cholinergic system damage in this study. Thus, we will carry out the improvement of EFDK against neurotoxicity and neuro-inflammation using amyloid beta1-42-induced mouse model.
[References]
1. Zhao, Wanyun, et al. Lycium barbarum Polysaccharides protect against Trimethyltin chloride-induced apoptosis via sonic hedgehog and PI3K/Akt signaling pathways in mouse neuro-2a cells. Oxidative medicine and cellular longevity, 2016, 2016.
2. Lee, Sueun, et al. Trimethyltin-induced hippocampal neurodegeneration: A mechanism-based review. Brain research bulletin, 2016, 125: 187-199.
3. Lattanzi, Wanda, et al. Gene expression profiling as a tool to investigate the molecular machinery activated during hippocampal neurodegeneration induced by trimethyltin (TMT) administration. International journal of molecular sciences, 2013, 14.8: 16817-16835.
4. Geloso, Maria Concetta et al., Trimethyltin-induced hippocampal degeneration as a tool to investigate neurodegenerative processes. Neurochemistry international, 2011, 58.7: 729-738.
7. In various article, the effect of TMT is described as pro-inflamatory and several NSAID agents, like indomethacin, are described to block the TMT effect. Maybe is the case also for the EFDK extract?
→ Thank you for your sincere suggestion. Nonsteroidal anti-inflammatory drugs (NSAIDs) you mentioned are a drug class that reduce pain, decrease fever, prevent blood clots and, in higher doses, decrease inflammation. NSAIDs work by inhibiting the activity of cyclooxygenase enzymes (COX-1 and/or COX-2). In cells, these enzymes are involved in the synthesis of key biological mediators, namely prostaglandins which are involved in inflammation, and thromboxanes which are involved in blood clotting.
→ According to another previous study, the mechanism of neurodegeneration after TMT injection appears to differ between the CA3 and CA1 regions in brain. COX-2 was endogenously expressed in the CA3 region in the normal control group, and the ratio of COX-2 for the pyramidal cell in the CA3 region was unchanged after TMT treatment. And the COX-2 inhibitor could not prevent the loss of pyramidal cells in this region. Metyrapone as a drug used in the diagnosis of adrenal insufficiency reduced the extent of TMT-induced damage to pyramidal cells in the CA1 region but not the CA3 region. These data exclude the possibility that the TMT-induced neuronal cell death in the CA3 region is related to a hypothalamic–pituitary–adrenal axis or an inflammatory process. However, no further mechanistic insights is yet available, and thus it remains to be further examined. This study suggested a novel finding that the COX-2-dependent pathway appears to assist TMT-induced neuronal degeneration in CA1 pyramidal cells but not in CA3 pyramidal cells in a corticosterone-independent manner. This COX-2 induction needs to be examined further since understanding the fundamental molecular mechanism underlying the temporospatial regulation of inflammatory pathways in the brain has become an increasingly important issue. Based on these studies, it is difficult to confirm the expression level of COX-2 in the TMT model, and it is difficult to confirm whether the effect of NSAIDs is adequately expressed.
However, our results suggest that EFDK regulates changes in the expression levels of p-Akt and NF-kB. Thus, it may indirectly control the expression level of COX-2 like the NSAIDs. Similarly, as mentioned above, when we examine the protective effect of EFDK in amyloid beta-induced mice in the near future, we will evaluate the expression level of COX-2, and examine the function of NSAIDs whether EFDK plays a role as NSAIDs as you mentioned.
[Reference]
1. Kim, Doo Kwun and Jang, Tae Jung. Cyclooxygenase‐2 expression and effect of celecoxib in flurothyl‐induced neonatal seizure. International journal of experimental pathology, 2006, 87.1: 73-78.
2. Tsutsumi, S., et al. Circulating corticosterone alters the rate of neuropathological and behavioral changes induced by trimethyltin in rats. Experimental neurology, 2002, 173.1: 86-94.
3. Koistinaho, J., et al. Expression of cyclooxygenase-2 mRNA after global ischemia is regulated by AMPA receptors and glucocorticoids. Stroke, 1999, 30.9: 1900-1906.
4. Shirakawa, Takafumi, et al. Temporospatial patterns of COX-2 expression and pyramidal cell degeneration in the rat hippocampus after trimethyltin administration. Neuroscience research, 2007, 59.2: 117-123.
Thanks for valuable suggestions!
We conclude the answer for your general review and detailed requirements above.
Also, if you have any additional supplements after checking, sincerely hope to modify.
Thank you for your attention.
